# Increased Enzymatic Activity of Acetylcholinesterase Indicates the Severity of the Sterile Inflammation and Predicts Patient Outcome following Traumatic Injury

**DOI:** 10.3390/biom13020267

**Published:** 2023-01-31

**Authors:** Aleksandar R. Zivkovic, Georgina M. Paul, Stefan Hofer, Karsten Schmidt, Thorsten Brenner, Markus A. Weigand, Sebastian O. Decker

**Affiliations:** 1Department of Anesthesiology, Heidelberg University Hospital, 69120 Heidelberg, Germany; 2Clinic for Anesthesiology, Intensive Care, Emergency Medicine I and Pain Therapy, Westpfalz Hospital, 67661 Kaiserslautern, Germany; 3Department of Anesthesiology and Intensive Care Medicine, University Hospital Essen, University of Duisburg-Essen, 45147 Essen, Germany

**Keywords:** trauma, cholinergic anti-inflammatory pathway, acetylcholine, point-of-care testing

## Abstract

Traumatic injury induces sterile inflammation, an immune response often associated with severe organ dysfunction. The cholinergic system acts as an anti-inflammatory in injured patients. Acetylcholinesterase (AChE), an enzyme responsible for the hydrolysis of acetylcholine, plays an essential role in controlling cholinergic activity. We hypothesized that a change in the AChE activity might indicate the severity of the traumatic injury. This study included 82 injured patients with an Injury Severity Score (ISS) of 4 or above and 40 individuals without injuries. Bedside-measured AChE was obtained on hospital arrival, followed by a second measurement 4–12 h later. C-reactive protein (CRP), white blood cell count (WBCC), and Sequential Organ Failure Assessment (SOFA) score were simultaneously collected. Injured patients showed an early and sustained increase in AChE activity. CRP remained unaffected at hospital admission and increased subsequently. Initially elevated WBCC recovered 4–12 h later. AChE activity directly correlated with the ISS and SOFA scores and predicted the length of ICU stay when measured at hospital admission. An early and sustained increase in AChE activity correlated with the injury severity and could predict the length of ICU stay in injured patients, rendering this assay a complementary diagnostic and prognostic tool at the hand of the attending clinician in the emergency unit.

## 1. Introduction

Systemic inflammation is an immediate immune response to noxious stimuli. This medical condition is often associated with rapid clinical deterioration and organ dysfunction, necessitating an immediate response. Early diagnosis of dysregulated systemic inflammation plays an essential role in the effective treatment and patient outcome [1].

Cholinergic activity has been shown to modulate the inflammatory response by inhibiting the pro-inflammatory activity and facilitating the release of anti-inflammatory cytokines [2,3]. Direct measurement of the cholinergic activity in vivo remains to be a challenge for clinicians. Various assays have been proposed to indirectly assess the cholinergic activity [4,5,6]. Indeed, measuring the activity of the acetylcholine hydrolyzing enzymes acetylcholinesterase (AChE) and butyrylcholinesterase (BChE) in blood has been proposed as a quick, simple, and readily available method for indirect assessment of the cholinergic activity in the body [7,8,9,10]. Previous studies of our group and others showed strongly reduced activity of the BChE associated with pathogen-induced [11,12,13] as well as sterile systemic inflammation [14,15,16,17,18,19,20,21]. The BChE enzymatic activity has been shown to correlate with the disease and injury severity in these patients. Most importantly, quick dynamics of the BChE enzyme activity during an ongoing inflammation renders this method prompt and efficient in an early diagnosis of emerging systemic inflammation. The rate of the BChE activity change enables a quick and sensitive method for the early detection of systemic inflammation [11,12,13,17,18,19].

AChE is an enzyme found predominantly in the neuromuscular synapses, cholinergic synapses of the central nervous system, and erythrocytes [22,23]. AChE is responsible for the rapid hydrolysis of synaptic acetylcholine. Moreover, erythrocyte-bound AChE can hydrolyze plasmatic (non-neuronal) acetylcholine and, therefore, modulate the cholinergic anti-inflammatory response [24,25,26,27]. Previous work of Kawashima and colleagues showed that components of the cholinergic system, including cholinergic receptors, choline transporters, and acetylcholine hydrolyzing enzymes (AChE and BChE), are expressed in certain immune cells (macrophages, T, and B cells) [28]. However, little is known about the physiological function of the extrasynaptic AChE and its role in the immune response.

We tested whether bedside-measured AChE activity in blood might be an early indicator of commencing inflammation. Furthermore, we tested whether a change in the AChE activity might predict patient outcome following traumatic injury.

## 2. Materials and Methods

### 2.1. Study Design

This was a secondary retrospective analysis of two prospective studies. The first, an observational clinical study, approved by the Ethics Committee of the Medical Faculty of Heidelberg (Trial-Code No. S-196/2014), was performed in the emergency room and surgical intensive care unit of the University Hospital of Heidelberg, Germany, between July 2014 and March 2015 [17].

The second, an observational cohort study, approved by the Ethics Committee of the Medical Faculty of Heidelberg (Trial-Code No. S-196/2014 and No. S-391/2015) and the Rhineland-Pfalz Medical Board (file number: 837.539.15/10307), was performed in the emergency room and intensive care unit of the BG Trauma Centre Ludwigshafen/Rhine, Germany, between April 2016 and December 2016 [19].

In both studies, informed consent was obtained from patients or their legal designees and all volunteers. The methods conformed to the relevant regulations and guidelines.

### 2.2. Subjects and Measurements

Patients were recruited from two previously described studies [17,19]. All patients had an Injury Severity Score of 4 or above, were treated promptly, and received goal-directed treatment. In both studies, AChE activity was measured using the point-of-care-testing (POCT) device ChE Check (Securetec Detektions-Systeme AG, Neubiberg, Germany; In-Vitro-Diagnostics Guideline 98/79/EG; DIN EN ISO 18113-2 and -3) from blood samples used for routine blood gas analysis, as previously described [11,17,19]. Conventional markers of inflammation (CRP and WBCC), along with the injury severity scores, have been obtained and recorded at the corresponding time points of these studies. Standardized protocols of the hospital laboratory were used to measure CRP (turbidimetry method) and WBCC (flow cytometry method). The severity of the disease was assessed with SOFA (Sequential Organ Failure Assessment) and ISS (Injury Severity Score) scores measured within the first 24 hours following hospital admission. SOFA is a clinical disease severity score consisting of 6 variables representing organ systems (respiratory, coagulation, hepatic, neurological, cardiovascular, and renal). Each organ system is assigned a value (0–4), comprising a SOFA score range from 0 to 24 [29]. ISS is a nominal scale based on body regions with a range from 1 to 75. Six body regions comprise the following score: head or neck, face, chest, abdominal or pelvic contents, extremities or pelvic girdle, and external. The severity of the injuries is scored by using Abbreviated Injury Score (AIS) ranging from 0 to 6. The ISS score is calculated as the sum of the squares of the highest AIS scores for the three most severely injured body regions [30].

The patients were grouped together for the retrospective analysis (*n* = 82). Time points were identified for the analysis of AChE activity: hospital admission (primary survey) and stabilization (secondary survey). The second measurement, corresponding with the stabilization of the patient (i.e., patient achieving a steady state, no immediate further procedures necessary), was defined between 4 and 12 h following admission. This time frame was chosen to fit measurement times from all data sets [31]. The median time between measurements was 720 (with an interquartile range of 360–720) min. The control group comprised a sample of 40 healthy volunteers with no sign of infection or evident chronic disease, where AChE activity was measured once after obtaining a signed consent (Table 1, Figure 1).

### 2.3. Statistical Analysis

The data from the two prospective studies were collated in an electronic database (Microsoft Excel for Microsoft 365 MSO, Version 2008, Microsoft Corp., Redmond, WA, USA). GraphPad Prism 9 (GraphPad Software, La Jolla, CA, USA, www.graphpad.com, accessed on 30 January 2023) was used to evaluate the data. D’Agostino and Pearson omnibus normality tests were used to verify the Gaussian distribution of the study groups. Data are presented as medians with an interquartile range (IQR). Statistical significance between the study groups was calculated by using the Wilcoxon matched-pairs signed rank test or Mann–Whitney test. The correlation was tested using the Spearman correlation test. Best-fit value was calculated by using a simple linear regression model. A *p*-value < 0.05 indicated statistical significance.

## 3. Results

We measured the activity of the acetylcholine hydrolyzing enzyme acetylcholinesterase in patients with traumatic injury at hospital admission (primary survey) and following initial treatment and stabilization (4–12 h following hospital admission, secondary survey). Measurements revealed significantly increased AChE activity in injured patients, as compared to those obtained from healthy volunteers (Figure 2a, Appendix A, Appendix A).

CRP activity measured at hospital admission remained unaffected and significantly increased in the second phase of the trauma treatment (Figure 2b, Appendix A). Initially, significantly increased white blood cell count (WBCC) recovered during the observation period (Figure 2c, Appendix A). 

We next evaluated whether the observed change in AChE activity correlated with the conventional inflammatory biomarkers CRP and WBCC. We performed the correlation analysis at the time point of hospital admission and at the stabilization phase, 4–12 h following hospital admission. AChE activity showed no correlation with unaffected CRP levels measured at hospital admission (Figure 3a). In contrast, AChE activity correlated well with the WBCC at this time point (Figure 3b). Next, a mild but significant correlation between AChE and CRP was observed at the time point of the secondary survey, 4–12 h following hospital admission (stabilization phase) (Figure 3c). However, a correlation between AChE and WBCC was lost at this time point (Figure 3d).

The discrepancy in the observed efficacy of the tested inflammatory biomarkers requires further examination. We thus examined whether conventional inflammatory biomarkers CRP and WBCC correlated with the severity of the disease, assessed by calculating SOFA and ISS scores. Correlation analysis was performed for the biomarkers obtained at the two time points: hospital admission and stabilization (secondary survey) following traumatic injury. Initially, unaffected CRP did not correlate with the disease severity score SOFA (Figure 4a) nor with the injury severity score ISS (Figure 4b) when measured at hospital admission. In contrast, WBCC showed a strong correlation with SOFA (Figure 4c) and ISS (Figure 4d) scores at the same time point. When measured during the stabilization phase, CRP strongly correlated with SOFA (Figure 4e) and ISS (Figure 4f) scores. Further analysis showed a loss of correlation between WBCC and SOFA (Figure 4g), as well as ISS (Figure 4h) during the stabilization phase of the trauma care.

We next tested whether a change in the activity of conventional inflammatory biomarkers, when measured at given time points, could predict the length of the patient’s stay in the ICU. CRP activity, when measured at hospital admission, did not correlate with the length of ICU stay (Figure 5a). Nevertheless, CRP activity obtained during the stabilization phase of the trauma care correlated well with the length of patient stay at the ICU (Figure 5b), suggesting that later CRP measurements better correlated with the ICU length of stay than those initially obtained from injured patients. Conversely, initially measured WBCC correlated well with the length of ICU stay (Figure 5c). WBCC obtained during the stabilization phase showed no correlation with the length of ICU stay (Figure 5d), indicating that only initially obtained, but not subsequent WBCC measurements, correlated with the length of the ICU stay.

Finally, we tested whether AChE activity, obtained at the hospital admission and during the stabilization phase of the trauma care, could be associated with the extent of organ dysfunction and with the severity of the traumatic injury. Indeed, AChE activity showed a moderate but sustained direct correlation with SOFA score (Figure 6a,b) during the observation time. Correlation analysis between AChE and the Injury Severity Score revealed comparable results: AChE activity, obtained both at the hospital admission and at a later time point (during the secondary survey), correlated significantly with the ISS (Figure 6c,d). Most importantly, AChE activity obtained from injured patients showed a sustained and significant direct correlation with the length of ICU stay (Figure 6e,f).

In addition, a confirmatory analysis of the disease severity scores revealed a strong direct correlation (SOFA vs. ISS, r = 81, Appendix A). Moreover, SOFA (Appendix A) and ISS score (Appendix A) strongly correlated with the length of the ICU stay.

## 4. Discussion

Our results show that an increase in AChE activity, an enzyme responsible for acetylcholine hydrolysis, could indicate a very early phase of injury-induced sterile inflammation. We observed that an increase in AChE activity following traumatic injury could predict the length of ICU stays for these patients, irrespective of the time of the measurement, rendering this method a reliable predictor for trauma patient outcome. Our results suggest that an early increase in AChE activity may precipitate cholinergic downregulation and support the proinflammatory phase of the early immune response.

A change in cholinergic activity, observed during sterile inflammation, has been shown to modulate immune reaction in the early phase of the inflammatory response [18,32]. Here we demonstrate that an increase in AChE activity, an enzyme responsible for acetylcholine hydrolysis, could indicate a very early phase of injury-induced sterile inflammation.

Interestingly, the observed change in AChE activity acts in an opposite manner to the activity pattern of the BChE, an analogous acetylcholine-hydrolyzing enzyme abundant in blood, showing strong activity reduction upon sterile inflammation [11]. The observed opposing activity pattern of the two acetylcholine-hydrolyzing enzymes upon the commencing sterile inflammation raises several questions. First, an increase in AChE activity is observed very early after traumatic injury. The observed increase persisted during the observation period and directly correlated with the injury severity. This finding indirectly suggests that the cholinergic activity might modulate the intensity of the initial immune response. Interestingly, previous research regarding the role of acetylcholinesterase in inflammation predominantly focused on the effects of pharmacologic enzyme inhibition following systemic inflammation and sepsis [33,34,35,36], as well as neuroinflammation and delirium [9,10,37,38,39,40,41,42,43,44]. In contrast, previous studies showed that BChE activity decreases upon sterile inflammatory challenge, peaking several hours later [17,19]. The question is raised: how do two enzymes complement each other to reduce inflammation? Both enzymes are shown to correlate with the severity of the injury, suggesting that the rate of the activity change of both enzymes corresponds to the intensity of the inflammatory response. Nevertheless, the end-effect of the two enzymes acting counteractively suggests that the cholinergic system might initially act pro-inflammatorily, which later turns into anti-inflammatory action. The alternating activity pattern of the acetylcholine hydrolyzing enzymes observed during injury-induced sterile inflammation might be associated with the previously described bi-phasic immune response following an inflammatory hit [45]. Indeed, proinflammatory properties of the cholinergic system have been described. Nevertheless, the site of cholinergic proinflammatory action has been limited to the respiratory system [46,47]. Mechanisms controlling this process remain unclear. A possible scenario upon sterile inflammation might be an initial downregulation of the cholinergic activity due to a rapid increase in AChE activity. Next, a feedback-coupled stimulation of AChE enzymatic activity might cause a decrease in BChE activity, an enzyme abundantly present in serum, which, in turn, would cause increased cholinergic (anti-inflammatory) activity. A sustained increase in the AChE/BChE activity ratio obtained from injured patients might, in part, support this hypothesis. To our best knowledge, the role of AChE in early sterile inflammation has not been previously addressed. Further studies are needed to test the proposed hypothesis. 

Furthermore, the observed increase in AChE activity following traumatic injury could predict the length of ICU stay of these patients, irrespective of the time of the measurement, rendering this method a reliable predictor for trauma patient outcome. Interestingly, our data showed that conventional inflammatory biomarkers CRP and WBCC did not persistently correlate with the severity of the disease. The correlation efficacy of these biomarkers strongly depended on the time point of the measurement. The inflammatory biomarker CRP showed an early activity increase following injury, which is in concordance with numerous studies rendering this protein a stable and sensitive biomarker in inflammation diagnostics [48,49,50]. Our study showed an increase in CRP activity starting six hours following injury, thus confirming the reports of previous studies describing peak CRP activity 24 h following inflammatory hit [51,52,53]. Nevertheless, CRP, described as an acute-phase protein, failed to indicate an inflammatory process when measured immediately after an injury. This observation might be explained by the latency time needed for protein synthesis in the body upon an inflammatory challenge [54,55]. Interestingly, a work from Richter and colleagues reported an interaction between CRP and the cholinergic system, rendering this protein an unconventional nicotinic alpha-7 agonist, protecting against the trauma-induced release of IL-1β cytokine during inflammation [56]. Our study could neither confirm nor deny this hypothesis; nevertheless, a possible interaction of CRP and the cholinergic system might not be excluded. Further studies addressing the neuro-immune interaction during sterile inflammation are required. 

White blood cells express components of the cholinergic system, including AChE [57,58,59,60]. Increased AChE activity may be associated with concurrent leukocytosis following traumatic injury. Further study is needed to test this presumption. Furthermore, white blood cell count showed an expected and significant early elevation upon traumatic injury. Nevertheless, the initially elevated cell count recovered after six hours, failing to predict the patient outcome at this time point. Our finding is in line with studies reporting a limited efficacy of WBCC in predicting patient outcomes [61,62,63]. Leukocytosis observed in trauma is predominantly a result of neutrophilia and not due to the release of immature cells or increased bone marrow production. This phenomenon has been shown to be rather short-lasting [64], which might partially explain the observed recovery trend and the deficiency of WBCC to predict the patient outcome at a later time point. Alternatively, extensive fluid therapy during resuscitation of injured patients might be responsible for reduced cell count at later time points [65]. 

The most surprising aspect of the data in this study is the predicting power of the bedside-measured AChE change, outperforming both conventional biomarkers by sustainably providing a consistent correlation with the injury and disease severity. This finding is in concordance with our previous studies reporting an early and sustainable negative correlation between BChE activity and length of ICU stay of patients with major traumatic injury [19], suggesting that monitoring of the cholinergic activity upon sterile inflammation might offer a complementary prognostic tool for the early decision-making process in the emergency department. Our data suggest that the efficacy of the conventional inflammatory biomarkers in predicting patient outcome after traumatic injury strongly depends on the time point of the measurement. Interestingly, the first measurement of AChE activity was increased in comparison with healthy individuals. This consistently correlated with the severity of the injury. Most importantly, our data suggest that measured AChE activity could predict the patient outcome when measured as early as upon hospital admission. 

Our study has several limitations. The number of included patients in this study is too low to confidently extrapolate conclusions to other clinical settings. This study does not address cholinergic activity upon pathogen-induced inflammatory response. Further multicentric studies addressing AChE activity during sterile and pathogen-induced inflammation might help to better understand the role of cholinergic activity in inflammation. Next, variable time intervals between the measurements might limit the interpretation of the results. This study did not include the analysis of immunomodulating agents (i.e., cytokines) nor a quantitative lymphocyte analysis. Next, the AChE, a modulator of cholinergic activity, plays an essential role in different systems (neuromuscular junction, synaptic activity of the cholinergic neurons in the central nervous system). The interaction of these systems with the activity of the AChE during sterile inflammation is beyond the scope of this study. Finally, the observed increase in AChE is rather subtle. A question regarding the clinical relevance of such a finding remains legitimate. We agree that the change in AChE activity, as a sole parameter, has a limited capacity to diagnose an inflammation. Nevertheless, if used in complement with the BChE, conventional inflammatory biomarkers, and clinical evaluation, AChE activity might help in the decision-making process at hospital admission. 

In summary, these findings contribute to the understanding of the role of cholinergic activity in the very early phase of sterile inflammation. In particular, the observed increase in the activity of acetylcholinesterase, an enzyme predominantly responsible for central cholinergic neurotransmission, sheds new light on the dynamics of the pathomechanisms underlying neuro-immune interactions during sterile inflammation. Furthermore, a very early and sustainable increase in the AChE activity may predict the length of ICU stay in patients following traumatic injury, rendering this assay a complementary diagnostic and prognostic tool at the hand of the attending clinician in the emergency unit.

## Figures and Tables

**Figure 1 biomolecules-13-00267-f001:**
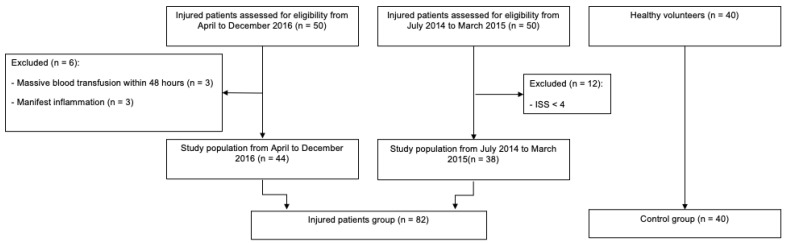
Flowchart of the patients and the volunteers included in this study.

**Figure 2 biomolecules-13-00267-f002:**
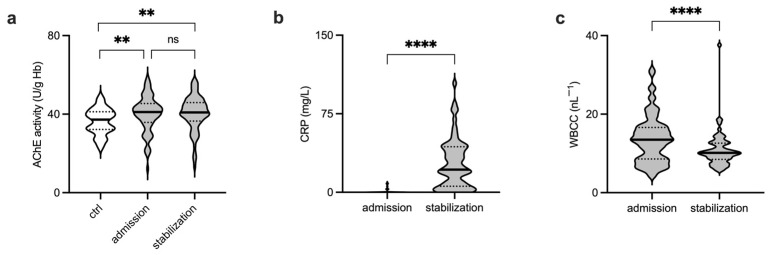
Significant increase in enzymatic activity of acetylcholine occurs during the initial period following traumatic injury. (**a**) Compared to measurements from healthy volunteers (ctrl), injured patients showed significantly elevated AChE activity at the hospital admission (Mann–Whitney test). AChE activity remained elevated during the stabilization phase (secondary survey, performed 4–12 h following admission and primary survey, Wilcoxon matched-pairs signed rank test). (**b**) CRP remained unaffected when initially measured in injured patients. A significant increase in CRP activity was observed during the stabilization phase. (**c**) Injured patients showed a significantly increased WBCC upon arrival to the hospital, followed by WBCC recovery during the observation period. Black horizontal lines in violin plots represent medians. Dotted black lines are quartiles. n.s.—not significant; AChE—acetylcholinesterase; CRP—C-reactive protein; WBCC—white blood cell count; ** *p* < 0.01; **** *p* < 0.0001.

**Figure 3 biomolecules-13-00267-f003:**
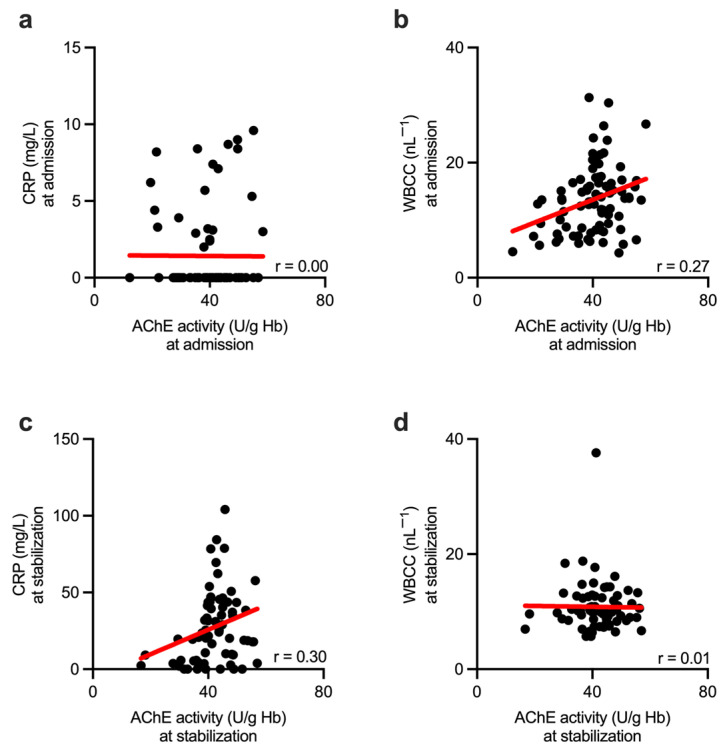
Conventional inflammatory biomarkers correlate with the AChE activity in a time-related manner. (**a**) AChE activity measured in injured patients at the hospital admission showed no correlation with (unaffected) CRP levels. (**b**) AChE activity correlated with WBCC when measured at the same time point. When measured at the time point of the secondary survey, 4–12 h following hospital admission (stabilization), AChE activity correlated with the CRP (**c**) but not WBCC (**d**). Dots represent single measurements. Red lines represent linear regression. AChE—acetylcholinesterase; CRP—C-reactive protein; WBCC—white blood cell count; r—Spearman correlation coefficient.

**Figure 4 biomolecules-13-00267-f004:**
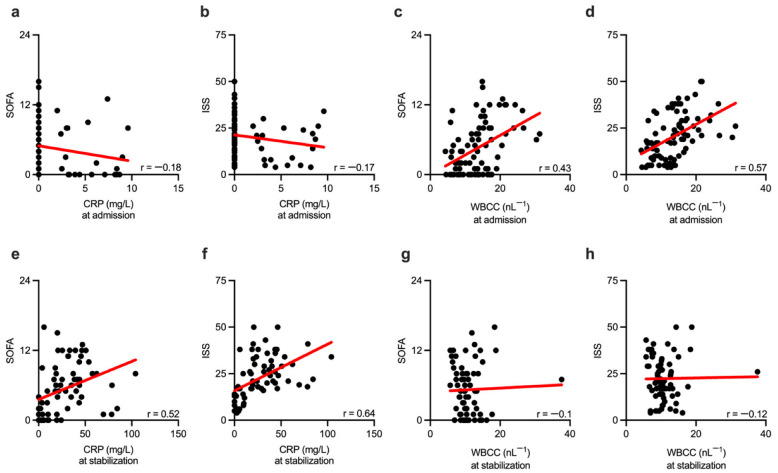
Correlation between measurement time and disease severity scores with conventional inflammatory biomarkers. CRP shows no correlation with the disease severity score SOFA. (**a**) nor with injury severity score ISS (**b**) when measured at the hospital admission. WBCC strongly correlates with SOFA (**c**) and ISS (**d**) at the same time point. Measurements obtained during the stabilization phase reveal a strong correlation between CRP and SOFA (**e**) as well as ISS (**f**) scores. WBCC showed no correlation with the disease severity scores SOFA (**g**) nor with ISS (**h**) at this time point. SOFA—Sequential Organ Failure Assessment; ISS—Injury Severity Score; CRP—C-reactive protein; WBCC—white blood cell count; r—Spearman correlation coefficient.

**Figure 5 biomolecules-13-00267-f005:**
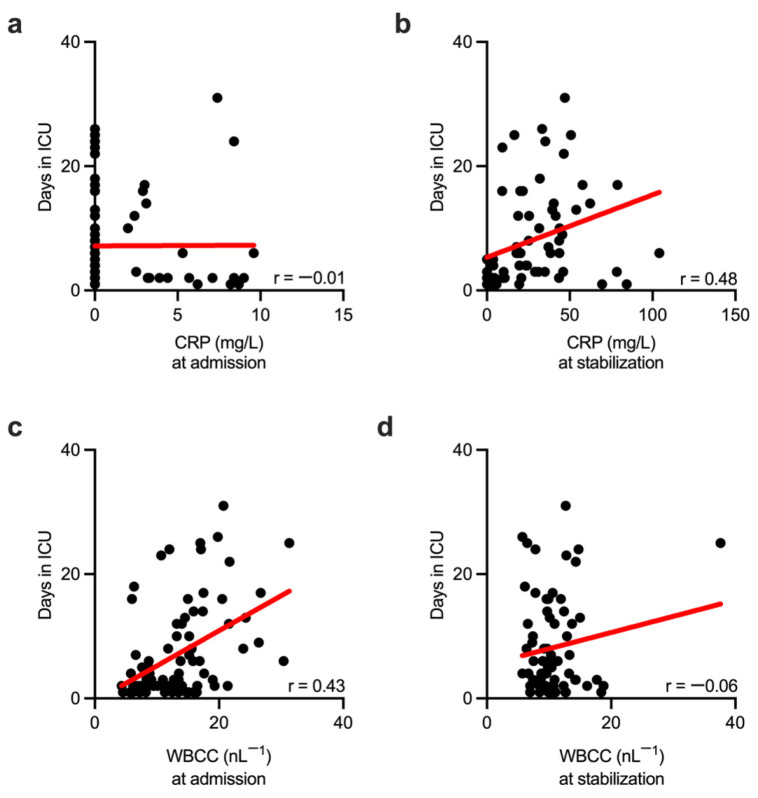
The efficacy of conventional biomarkers in predicting patient outcomes depends on measurement time. Scatter plots show no correlation between length of ICU stay and CRP activity measured in injured patients at hospital admission. (**a**) CRP measured during the stabilization phase (secondary survey) strongly correlated with the length of the ICU stay in these patients (**b**) WBCC obtained at admission correlated well with the length of the ICU stay (**c**) WBCC obtained during the stabilization phase failed to correlate with the length of ICU stay (**d**) ICU—intensive care unit; CRP—C-reactive protein; WBCC—white blood cell count; r—Spearman correlation coefficient.

**Figure 6 biomolecules-13-00267-f006:**
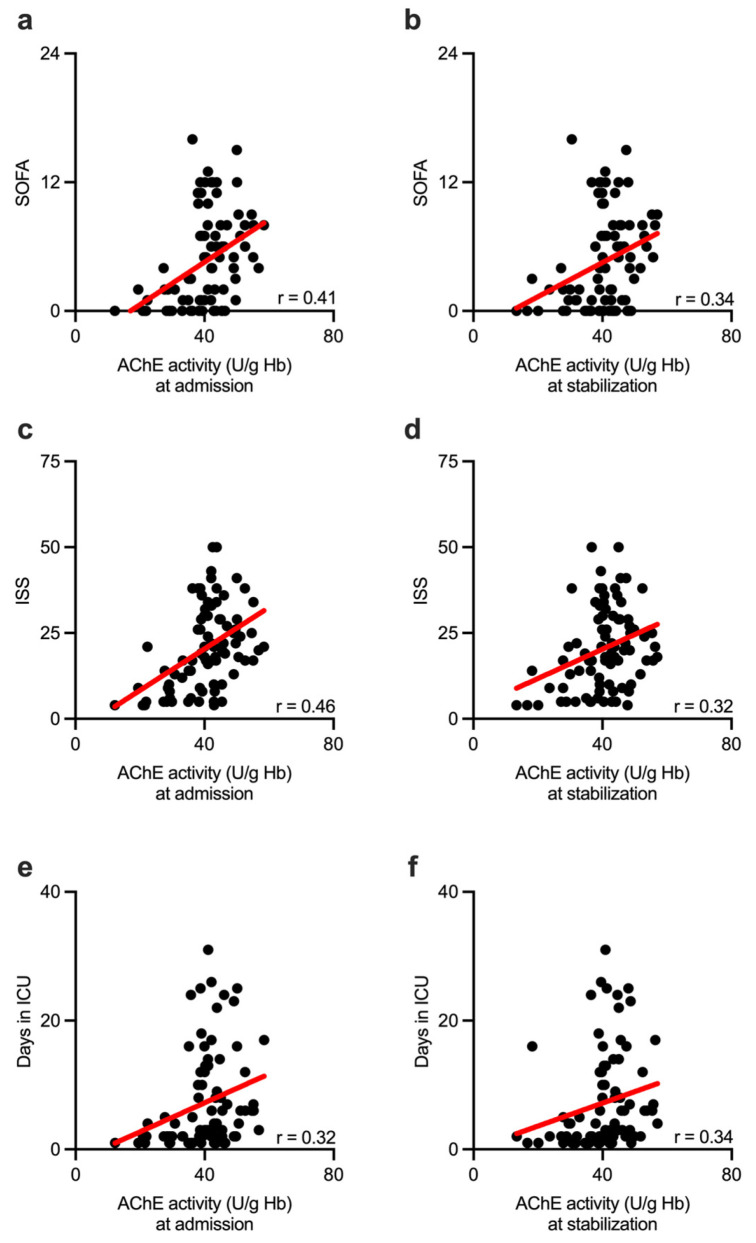
AChE activity correlated with the disease severity scores and the length of the ICU stay throughout the early period of the sterile inflammation. AChE activity was measured at hospital admission (**a**) and at the stabilization time point, 4–12 h following hospital admission (**b**), correlated with SOFA score obtained from injured patients 24 h following hospital admission. AChE activity was measured at hospital admission (**c**) and, following the stabilization (**d**), correlated with the ISS score obtained from injured patients. Initially measured AChE activity (**e**), as well as AChE activity obtained during the stabilization phase (**f**), correlated with the length of ICU stay of injured patients. AChE—acetylcholinesterase; SOFA—Sequential Organ Failure Assessment; ISS—Injury Severity Score; ICU—intensive care unit; r—Spearman correlation coefficient.

**Table 1 biomolecules-13-00267-t001:** Basic demographic data of volunteers as well as demographics and clinical scoring data of the injured patients. * Median with interquartile range.

Volunteers	
Number of volunteers	40
Age of volunteers (years) *	38 (29–54)
Gender of volunteers (m/f)	23/17 (58%/42%)
Trauma Patients	
Number of trauma patients	82
Age of trauma patients (years) *	53 (33–69)
Gender of trauma patients (m/f)	55/27 (67%/33%)
SOFA *	4 (0–8)
Days in ICU *	4 (2–12)
Traumatic injury	
ISS *	19 (10–29)
ISS 4–15 (number of patients)	29
ISS 16–24 (number of patients)	24
ISS > 25 (number of patients)	29
ISS body regions	
Head or neck (number of patients)	48
Face (number of patients)	16
Chest (number of patients)	55
Abdomen or pelvic contents (number of patients)	23
Extremities or pelvic girdle (number of patients)	43
External (number of patients)	30

## Data Availability

The data presented in this study are available on request from the corresponding authors.

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
