# Peer review of "Increased Enzymatic Activity of Acetylcholinesterase Indicates the Severity of the Sterile Inflammation and Predicts Patient Outcome following Traumatic Injury"

_biomolecules, 2023, doi:10.3390/biom13020267_

Round 1
Reviewer 1 Report
This secondary retrospective analysis from two observational studies investigated AChE and BChE activity among injured and healthy volunteers between 2014 and 2017. The outcomes investigated included the association of bedside-measured AChE activity in blood at admission and stabilization with WBCC, CRP, ISS, SOFA, and days in ICU. The role of the cholinergic system in critical and traumatic injury has been poorly described due to previous limitations with sample collection that the investigators address by using the bedside POCT device. The authors found that injured patients had increase in AChE initially and sustained, which correlated with injury severity. Overall, the results are intriguing and useful in generating further hypothesis while also generating useful early data.
Major critiques:
1. The hypothesis listed – that non-neuronal cholinergic activity requires AChE to adequately modulate inflammatory response – may be true was not actually tested in this study. Please clearly define primary and secondary hypotheses (or outcomes) within final paragraph of the Introduction that were assessed regarding AChE levels in trauma and association with inflammation, outcomes.
2. Within Materials and Methods, Study Design, please describe the recruitment and inclusion of healthy volunteers.
3. Within Materials and Methods, Study Design, please specify how stabilization time point was defined. What was the median time from admission to collection and admission to stabilization?
4. Please provide further explanation of the scientific/physiologic rationale behind using AChE/BChE activity ratio since these enzymes are found in different locations and have different purposes in the body. Furthermore, could the differential directions of the enzymes with injury be due to the differential location/roles as opposed to pro-inflam then anti-inflam responses?
5. Within Results sections, please refrain from stating conclusions or adding discussion points. These should be made within the Discussion section. The following examples are included:
· P6L175: “As expected,”
· P6181-2: “These results support a presumption…”
· P6L194-6: “These data suggest a causality…”
· P8L234-6: “These results are significant in…”
· P9L247: “we hypothesized”
· P10L270-2: “These results further support…”
Minor critiques:
1. The first paragraph of the Introduction describes systemic inflammation, which is an adaptive and often beneficial response. The cause of injury is frequently attributed to a dysregulated, or exaggerated, inflammatory responses. The authors may clarify “excessive systemic inflammation” with one of these more commonly used terms (dysregulated/exaggerated).
2. Within the Introduction P2L50, please remove “could”.
3. Within the Introduction P2L52, please remove “Moreover”.
4. Within the Introduction P2L62, please remove “Moreover”.
5. Within the Introduction P2L68, please remove “Here”.
6. Within Materials and Methods, Subjects and Measurements, please be more careful with the language and description of the participants. It is stated that “All patients had an Injury Severity Score of 4 or above”, however the study also included health volunteers and patients with ISS < 4 were considered “controls”. Later in the section it is also noted that “an ISS of three or below” was included within the study.
7. Within Materials and Methods, Subjects and Measurements, please spell out “POCT” if it is an abbreviation.
8. Please describe the Discussion by primary hypothesis or outcome first followed by discussion of secondary hypotheses/outcomes.
9. Within the final sentence, the suggestion that these preliminary results “could effectively predict” outcomes is a bit out of scope for the study results. Would recommend editing to “may predict”.
Reviewer 2 Report
Comments for authors
The authors tested whether bedside-measured AChE activity in blood might indicate a change in the cholinergic activity following the inflammatory challenge. They hypothesized that the non-neuronal cholinergic activity requires AChE to adequately modulate the inflammatory response. Also, the authors said that the role of AChE in early sterile inflammation has not been previously examined.
Introduction.
Line 60 and line 63. Both sentences started with Moreover, please edit one of them.
Methods:
Line 89. Please describe how you measured the activities of AChE and BChE or provide a reference. Do AChE is measured in erythrocytes and BChE in plasma? Also, provide references or methods for other examined parameters (CRP, WBC).
Line 113. The authors say that a control group contained healthy volunteers (n=40) plus 12 patients who were recruited with an ISS of three or below and who are considered mildly injured or not injured. However, in my opinion only not injured subjects may be a control group. Please, remove all the injured patients from the control group and then do statistical analyses.
Line 135. The authors said: Measurements revealed increased AChE activity in injured patients, as compared to those obtained from healthy volunteers (Figure 2a, Suppl. Figure 1a, Suppl. Table 1). Do these differences were significant? In the table and figure it is not marked. If these differences are not significant, can ACHE be an efficient predictor of the length of ICU stay in patients following traumatic injury, as the authors said later? Also, if AChE activity is staying stable for 6-12 hours following hospital admission, as shown in supplementary figure 1a then changes in AChE/BChE ratio depend only on the changes in activity BChE. Please, explain these results better.
Figure 2. It is not clear whether the measured parameters differ significantly between healthy volunteers and patients as well as if have differences between admission and stabilization. If there are any significant differences, mark them in figures 2a,2b,2c, and 2d, and in line with it write the results and discussion.
Supplementary Table 1 and Supplementary Figure 1. It is not necessary to present the same result in two ways. Decide whether you will display the activity of AChE and BChE as a table or as a figure. Also, give titles to tables and figures in supplementary materials.
Round 2
Reviewer 2 Report
Please, the correct number of patients who were included in the control group in Figure 1. (Flowchart of the patients and the volunteers included in the study).
In part Results, when the authors describe their results shown in Figure 2, need to add the word significantly for all parameters which had significant changes.
Author Response
Dear Editor,
Dear Reviewer 2,
Thank you for giving us the opportunity to revise and improve our manuscript. We addressed two minor points raised by the Reviewer.
Although our manuscript has been uploaded with the revised Figure 1, we noticed that the original Figure 1 remained in the pdf version of the submitted revised version of our manuscript. We apologise for this technical mishap. The latest version of the manuscript contains the new and revised Figure 1 with corrected number of volunteers.
We added the word "significantly" to all reported results of the Figure 2, where applicable. The changes included the Results section as well as Figure legend of the Figure 2.
Changes can be found in lines: 132, 135, 137, 140, 142, 146 and 148.
We hope that you will find the revised manuscript suitable for publication in the Biomolecules.
Sincerely,
Aleksandar Zivkovic